# The Influence of Daily Temperature Fluctuation on the Efficacy of Bioinsecticides on Spotted Wing Drosophila Larvae

**DOI:** 10.3390/insects14010043

**Published:** 2022-12-31

**Authors:** Maristella Mastore, Silvia Quadroni, Alberto Rezzonico, Maurizio Francesco Brivio

**Affiliations:** 1Laboratory of Environmental Entomology and Parasitology, Department of Theoretical and Applied Sciences, University of Insubria, 21100 Varese, Italy; 2Laboratory of Ecology, Department of Theoretical and Applied Sciences, University of Insubria, 21100 Varese, Italy

**Keywords:** invasive species, global warming, thermal stress, *Drosophila suzukii*, entomopathogen, pest management

## Abstract

**Simple Summary:**

The main purpose of this work was to evaluate the possible influence of daily temperature variations on the interaction between entomopathogens and the target insect *Drosophila suzukii*. *Drosophila suzukii* is an invasive alien species whose spread in non-native areas has increased in recent years due to global warming and which could be controlled by bioinsecticides such as nematodes and bacteria. The effectiveness of both nematodes and bacteria was evaluated by interaction assays performed under controlled conditions. The selected temperature ranges simulated the day/night temperature fluctuations detected during the main breeding periods of this dipteran in temperate regions. The results obtained show that the influence of temperature fluctuations is rather drastic on nematodes, while the efficacy of bacteria does not change significantly. From an application point of view, studying the effects of environmental temperature changes on entomopathogens used in biological control is extremely important to improve their effectiveness in controlling insect pests.

**Abstract:**

Global climate change is allowing the invasion of insect pests into new areas without natural competitors and/or predators. The dipteran *Drosophila suzukii* has invaded both the Americas and Europe, becoming a serious problem for fruit crops. Control methods for this pest are still based on the use of pesticides, but less invasive and more sustainable methods, such as biocontrol, are needed. Variations in environmental conditions can affect the efficacy of bioinsecticides influencing their behavior and physiology besides that of the target insects. In this work, we developed a system that simulates the daily temperature fluctuations (DTFs) detected in the environment, with the aim of studying the influence of temperature on biocontrol processes. We investigated the effects of DTFs on the efficacy of four bioinsecticides. Results showed that DTFs modify the efficacy of some entomopathogens while they are ineffective on others. Specifically, the bacterium *Bacillus thuringiensis* is the most effective bioinsecticide under all conditions tested, i.e., low DTF (11–22 °C) and high DTF (17–33 °C) compared to constant temperature (25 °C). In contrast, nematodes are more sensitive to changes in temperature: *Steinernema carpocapsae* loses efficacy at low DTF, while *Steinernema feltiae* and *Heterorhabditis bacteriophora* are not effective in controlling the target dipteran. This work provides a basis for reviewing biological control methods against invasive species in the current context of climate change.

## 1. Introduction

The massive increase in greenhouse gas emissions in the atmosphere, due to both natural factors and human activity, is the main cause of global warming and related drastic climate changes [1]. This has led to an increase in the number of warm days and nights, extreme weather events, and sea levels. Moreover, higher evaporation due to global warming has also led to a rise in overall precipitation, even if some areas have become drier due to region-specific changes [2]. Such alterations in environmental conditions can severely affect the behavior and physiology of plant and animal species in ecosystems [3]. Particularly, insects are poikilothermic organisms, and temperature alterations can influence their behavior by increasing their activity, expanding their geographical distribution, and increasing their survival rate and the number of generations per year [4,5,6]. In the case of insect pests, this can lead to extensive damage to plants and crops in newly colonized areas [7,8,9]. Linking changes in climate to interspecific interactions in natural communities and agroecosystems is extremely important in controlling the spread of phytophagous insects [5,10,11].

The impact of climate change on agroecosystems, and in particular, the effects of heat waves, are important issues highlighted by the global scientific community [5]. These seasonal and long-term changes affect the population dynamics of pests, the presence and effectiveness of natural enemies, and, consequently, the effectiveness of crop protection technologies. It is well known that environmental factors have both direct and indirect impacts on biological control agents by modulation of their life cycle, rate of development, survival, fecundity, parasitism, and dispersal [9,12,13]. Since the effectiveness of biological control systems depends largely on the interactions between entomopathogens and insect pests, the knowledge about how these relationships are affected by climate variations is of crucial importance for understanding and improving pest management under the global climate change scenario. Thus, the goals of future research on the effects of climate change on insect pests should include a thorough review of integrated pest management strategies, besides the monitoring of climate and pest populations and the use of modeling prediction tools.

In addition to being a problem for biodiversity and agriculture, pest spread has increased the use of pesticides, which represent a major environmental and health risk factor. Their adverse effects are correlated to various diseases, such as dermatological, gastrointestinal, neurological, carcinogenic, respiratory, reproductive, and endocrine damages. Moreover, the fate, distribution, and persistence of pesticides in environmental compartments are affected by climate change [14,15,16,17]. Due to the serious issues related to the use of pesticides, in recent years, the application of bioinsecticides in biological control procedures is increasing thanks to their safety and low environmental impact [18,19]. In biological control, bioinsecticides, including nematodes and microorganisms such as fungi, bacteria, and viruses, as well as predators such as parasitoid wasps, are normally used. The commercial formulations of bioinsecticides report indications on the application methods ensuring their efficacy, such as substrate typology, targets, and optimal temperatures of use. However, for proper selection of the most suitable bioinsecticide, it is important to consider the physiological conditions of both the bioinsecticide and the target, as they can be sensitive to daily temperature fluctuations (DTFs).

The effects of DTFs on insect physiology and their interactions with entomopathogens are still understudied [20,21,22,23] compared to a large amount of research carried out on the effects of constant temperature (CT) [24,25,26,27]. Since DTFs more closely represent the situation in which most organisms perform their life cycle and have been shown to influence the activity, development, and reproduction of ectothermic organisms [8,26,28], it is crucial to consider this aspect when planning biocontrol trials carried out both in the laboratory and in the greenhouse. Considering the DTF influence on the bioecology and reproductive biology of biocontrol agents (both endo- and ectoparasites) under laboratory-controlled conditions is indeed critical for eventual subsequent field releases [23].

One of the pests of great agroeconomic impact is *Drosophila suzukii*, also named Spotted Wing Drosophila (SWD). This dipteran, native to Southeast Asia, has invaded North and South America and Europe, causing severe crop losses of soft thin-skinned fruits [29]. SWD females lay eggs in the pulp of healthy ripening fruits by means of an ovipositor; this reproductive process results in high crop losses due to larval feeding and because ovipositor injury can promote secondary infections by phytopathogens [30]. Changes in environmental temperature can affect the oviposition, development, and growth rates of SWD populations [31,32,33,34]. Control of SWD has relied mainly on the use of chemical insecticides [35], but the rapid reproduction rate of dipterans requires multiple administrations, which can cause a significant increase in residues in fruit and the emergence of resistance forms [36]. However, as suggested by current regulations provided by international organizations, such as the European Environmental Agency, the European and Mediterranean Plant Protection Organization, and the World Health Organization [37,38,39], it is advisable to avoid chemical control interventions by replacing or complementing them with biological control methods.

Among the most commonly used bioinsecticides in SWD control are entomopathogenic nematodes (EPNs) and microorganisms such as *Bacillus thuringiensis* [35,40,41,42], thus our purpose was to investigate the efficacy of the nematodes *Steinernema carpocapsae* (Sc), *Steinernema feltiae* (Sf), *Heterorhabditis bacteriophora* (Hb), and the bacterium *B. thuringiensis* var. *kurstaki* (Btk), on the larval stages of SWD, under DTF conditions detected in two periods of the year (late spring and summer), corresponding to the reproductive window of SWD in temperate regions such as north and central Italy, where dramatic colonization of agricultural areas with large crops of thin-skinned fruits was detected [43].

Although there are many factors that influence the reproductive cycle of SWD [44] and bioinsecticide efficacy, they are believed to be closely related to climatic conditions [45], thus, to temperature variations. However, most of the relevant literature on SWD and bioinsecticides refers to studies performed at CT. To our knowledge, only a few recent studies have been carried out under DTF conditions but testing the efficacy of chemical pesticides on other insect species [17,46]. Thus, in our work, we first address the issue of thermal stress induced by DTFs in the biological control of SWD carried out by bioinsecticides.

From the results obtained, it is possible to postulate an effective influence of thermal variations on both bioinsecticides, particularly nematodes, and target insects; these preliminary data obtained in the laboratory provide the first indications for a revaluation of the applicability of these organisms in biological control. We are aware that to achieve a more comprehensive picture of the effects of climate change on biocontrol processes, further investigations are needed, particularly with different temperature ranges, tested on either other insect species or bioinsecticides.

## 2. Materials and Methods

### 2.1. Chemicals and Instruments

All reagents were supplied by Sigma Chemicals (St. Louis, MO, USA) and Merck Millipore Ltd. (Tullagreen, Cork, Ireland). Instruments were supplied by Bio-Rad Laboratories (Detroit, MI, USA) and Celbio Spa (Milan, Italy, EU), and Snijders Labs (Tilburg, Netherlands). Centrifugations were carried out with a SIGMA 1–14 (SciQuip Ltd., Newtown, Wem, Shropshire, UK) microcentrifuge and an Eppendorf 5804R (Eppendorf, AG, Hamburg, Germany) centrifuge. All materials, buffers, and solutions were autoclaved or filtered by 0.22 µm Minisart filters (Sartorius, Goettingen, Germany). For microscopy observations, a stereomicroscope SZQ4 (OPTIKA Italy Srl, Ponteranica (BG), Italy) connected to an OPTIKA mod. A C-HP digital camera was used.

### 2.2. Drosophila suzukii Stocks and Bioinsecticides

The third stage of *D. suzukii* larvae used for all assays was obtained from a laboratory culture of specimens collected in Catalonia (NE Spain), kindly provided by Prof. Del Pino (Autonomous University of Barcelona, UAB) in 2018. SWD larvae were reared on a specific diet (10.2 g agar, 50 g sugar, 60 g cornmeal, 10 g soybean four, 60 g yeast, 3 g Nipagin, 10 mL ethanol, and 5 mL propionic acid in 1 L of tap water) and maintained in a climatic chamber at 25 °C, 45% relative humidity (RH), with a photoperiod of 12:12 h. Only healthy larvae at early growth stages (L1-L2) were used for each assay.

The efficacy of bioinsecticides on SWD larvae at CT or DTF conditions was assessed by administering different concentrations of Btk (strain EG 2348) or EPNs, i.e., Sc, Sf, and Hb, supplied by Bioplanet (Bioplanet Srl, Italy). All EPNs species were provided at the third Infective Juvenile (IJ) stage, dehydrated in clay granules, and stored at 8 °C to retain the cryptobiotic state. For the assays, inert material was removed; 3 g of nematodes were dissolved in 100 mL of sterile dechlorinated water, then nematode suspension was layered on a discontinuous sucrose gradient (75–50–25%) and centrifuged at 110 ×*g*, for 10 min, at room temperature. Nematodes were recovered at the 25–50% sucrose interface, washed three times in dechlorinated sterile tap water to remove contaminants, and resuspended in tap water. The number of nematodes used in the assays, their viability, and their presence inside dead larvae were evaluated by stereomicroscopy. Btk was grown in a culture medium at 30 °C overnight, under shaking; spores and crystals were isolated as described by Cossentine et al. [47] and observed by phase-contrast microscopy. Finally, spores and crystals were lyophilized and, before use, resuspended in sterile filtered tap water.

### 2.3. Biocontrol Assays

To assess the lethality of bioinsecticides on SWD larvae, the agar trap method was used; the substrate for infection consisted of a thin layer of 0.75% soft agar mixed with 5% sucrose in sterile tap water; the solution was heated to 100 °C and then poured (4 mL) into Petri dishes (Ø 5 cm). The substrate was sufficiently transparent to allow the easy and accurate evaluation of larval morphology and viability. For tests with Btk, the agar solution was cooled to 37 °C, and afterward, a suspension containing Btk spores and crystals was added. For assays with nematodes, 1 mL of EPN suspension was added to the trap after the solidification of the agar. Twenty SWD larvae were transferred into the agar trap and incubated under different temperature conditions; all trials were carried out with six replicates. The climatic chamber used for the assays was set with two different thermal ranges, from 11 to 22 °C named low DTF (DTF_L_), and from 17 to 33 °C named high DTF (DTF_H_); the chamber modified the temperature by gradually distributing the change over 6 h, until the set temperature was reached (Appendix A). In addition, constant-temperature (CT, 25 °C) assays were performed for comparison. Since *D. suzukii* larvae develop inside the pulp of thin-skinned fruits, in all assays, RH in climatic chambers was kept constant at about 45%, allowing to keep the percentage of agar substrate wetness in the traps at about 90%, i.e., the average of RH values measured with a Data Logger Recorder, Elitech GSP-6 (Elitech, London, UK) within the pulp of different fruit varieties (RH (%) ± SD in grapes: 96 ± 0.8, blueberries: 92 ± 1.1, raspberries: 86 ± 1.3, strawberries: 87 ± 0.7). The assays with Btk were carried out using five increasing concentrations (International Units/mL): 640, 1900, 3800, 7600, and 15,200 IU/mL. All EPNs species were administered at 1 × 10^2^, 2 × 10^2^, 4 × 10^2^, 8 × 10^2^, and 1.6 × 10^3^ IJ/mL. The mortality of SWD larvae was monitored under a stereomicroscope at 24 and 48 h after incubation with the bioinsecticides. Larvae lacking movement were considered dead. All controls were created by incubating SWD larvae with the same volume of sterile tap water without bioinsecticides. Considering the optimal temperature range recommended by the manufacturers to ensure EPN efficacy, i.e., 14–26 °C for Sf, 14–35 °C for Sc, and 19–33 °C for Hb (Koppert BV, https://www.koppert.com/, accessed on 15 May 2022), we avoided performing assays at DTF_H_ for Sf, and DTF_L_ for Hb.

### 2.4. Data Analyses

For each bioinsecticide (i.e., Btk, Sc, Sf, Hb) administered at different temperature conditions (i.e., CT, DTF_L_, and/or DTF_H_), we showed the percentage of SWD larvae mortality detected at different concentrations (i.e., the control plus five increasing concentrations) and times (i.e., 24 and 48 h), as mean plus standard deviation of six replicates. We applied logistic regression (logit model), considering temperature conditions, concentrations, and times as categorical explanatory variables and the proportions of dead larvae (out of the 20 larvae tested in each replicate) as response variables [48]. After the regression, pairwise comparisons were carried out between estimated marginal means to detect possible significant (*p* < 0.05) differences in insect mortality between different temperature conditions at the same concentration and time, between different concentrations at the same time and temperature condition, and between times at the same concentration and temperature condition. Moreover, we performed Probit analysis with 95% confidence intervals when the SWD mortality range found by testing the different selected concentrations of bioinsecticide was suitable for calculating lethal concentrations (LC_50_ and LC_90_). Finally, we applied logistic regression also to compare the efficacy of different bioinsecticides (inversely related to SWD mortality) at 48 h at the highest concentration and the different temperature conditions assessed. All statistical analyses were performed using XLSTAT v. 2021 and R v. 4.2.2 (stats and emmeans package) [49] software.

## 3. Results

### 3.1. Effects of DTF on B. thuringiensis Administration

Agar trap assays were carried out with Btk at different increasing concentrations (from 640 to 15,200 IU/mL), and SWD larvae mortality was assessed at 24 and 48 h, at two ranges of DTF, 11–22 °C (DTF_L_) and 17–33 °C (DTF_H_), and constant temperature (CT, 25 °C) (Figure 1). SWD mortality significantly differed for all three considered factors (i.e., temperature, concentration, and time), and the interaction between each couple of factors and between all three factors was significant (Table 1).

For each temperature condition and time, most of the comparisons between the concentrations evaluated were significant, and, in all cases, the higher the concentration, the higher the SWD mortality (Appendix A). Although the average values of SWD mortality were always higher at 48 h than at 24 h, most comparisons in DTF conditions were not significant (Appendix A).

Except for the lowest concentration at 24 h between CT and DTF_H_ and the highest concentration at 48 h between CT and DTF_L_, no significant differences were detected between the different temperature conditions (Appendix A). However, LC_50_ at 48 h was lower at DTF_H_ (3915 IU/mL) than at CT (4876 IU/mL) and DTF_L_ (6768 IU/mL), showing a higher efficacy of the bioinsecticide at this temperature range. LC_90_ values were similar between DTF_H_ and CT while higher for DTF_L_ (Appendix A). At the colder temperature range, the average SWD mortality exceeded 50% at 7600 IU/mL instead of 3800 IU/mL for CT and DTF_H_ and remained below 90% at the highest concentration, reaching instead 92.5% and 97.5% at DTF_H_ and CT, respectively.

### 3.2. Effects of DTF on S. carpocapsae Administration

SWD larvae were co-incubated in agar traps with various amounts of nematodes (from 100 to 1600 IJ/mL) at the three temperature conditions, and the mortality of SWD larvae was monitored at 24 and 48 h after incubation (Figure 2).

As with Btk, in this case, the SWD mortality significantly differed for all three considered factors, and the interaction among each couple of factors and between all three factors was significant (Table 1). A significant difference in the efficacy of Sc at different temperature conditions was evident, mainly for the highest concentrations (Appendix A). The effects of temperature fluctuation, although in a range suitable for the action of the parasite, reduced its lethality when compared to the constant temperature. In fact, in the CT assay, at the highest concentration, SWD mortality reached 85.8% at 24 h after incubation and 92.5% at 48 h. At the same time, in the DTF_L_ condition, SWD mortality did not exceed 17.5 and 33.3%, respectively. The patterns observed for DTF_H_ were similar instead to those of CT, even if some significant differences, at the lowest concentration at 48 h and at the highest concentration at both times, were detected (Appendix A). However, for both temperature conditions, most of the comparisons among the concentrations evaluated were significant, with an increasing SWD mortality at increasing concentrations (Appendix A). In the case of DTF_L_ condition, the mortality detected at all the concentrations differed only from the control, except for the two lowest concentrations versus the maximum concentration at 48 h (Appendix A). Correspondingly, significant differences in the SWD mortality were observed between 24 and 48 h for DTF_H_ and, in some cases (800 and 1600 IJ/mL) for CT, while only at the maximum concentration for DTF_L_ (Appendix A).

### 3.3. Effects of DTF on S. feltiae Administration

The administration of Sf showed a low efficacy at both the temperature conditions evaluated (Figure 3).

In the CT assay, SWD mortality did not exceed 43.3% at the highest concentration 48 h after incubation. A further significant decrease in efficacy was found in the DTF_L_ assay at 48 h (Table 1 and Appendix A). In this case, the maximum SWD mortality was 25.8%. However, at both temperature conditions, all the tested concentrations caused significantly different SWD mortality compared to the control, and also significant differences were detected among the two highest concentrations and the lowest one (Appendix A). Significant differences in SWD mortality between 24 and 48 h were only detected at CT (Appendix A).

### 3.4. Effects of DTF on H. bacteriophora Administration

The results of the Hb administration were comparable to those of Sf. The efficacy against the SWD larvae was low at both the temperature conditions evaluated (Figure 4).

The maximum average SWD mortality detected in this study was 24.1% at the highest concentration at 48 h in the DTF_H_ condition. Although the logistic regression results were significant for all three considered factors and their relative interactions (Table 1), only a few pairwise comparisons were significant (Appendix A).

### 3.5. Comparison of the Maximum Efficacy of Bioinsecticides

Figure 5 shows a comparison of the results obtained with the entomopathogens at the highest concentration assessed (15,200 IU/mL for Btk and 1600 IJ/mL for nematodes), 48 h post administration (DF = 9, χ^2^ = 320, *p* < 0.0001). Unlike nematodes, Btk was particularly effective against SWD larvae and was only slightly affected by DTFs. Sc showed high activity against the dipteran but was sensitive to DTFs, particularly in the lower range (DTF_L_, 11–22 °C). Both Sf and Hb had low activity both at CT and DTF conditions.

## 4. Discussion

In this study, we considered the issue of daily temperature fluctuations (DTFs) recently detected in temperate areas where a wide invasion of the alien insect *D. suzukii* has been observed [43]. Both the significant increase in temperature and the wider thermal ranges have increased the development and spread of this insect pest [50,51]. However, the knowledge of the effects of different thermal conditions on SWD biology, extremely important for determining the presence and risk of infestation of this dipteran and for developing more effective biocontrol strategies, is still scarce [31].

Studies on SWD control should certainly be carried out in the field by means of various systems, such as nets, mass trapping, protective cloths, mulching, and natural compounds [52], and by evaluating the introduction to orchards of various entomopathogens and hymenopteran parasitoids [53,54]; but, although these studies are fundamental, they do not allow for an in-depth investigation of possible temperature variations, which can only be achieved by extensive laboratory trials under strictly controlled conditions. In biological control assays, careful control of environmental conditions can be carried out in climatic chambers, which allow constant and controlled variation of parameters, thus providing a comprehensive picture of the possible physiological changes to which bioinsecticides and their target insects are susceptible.

In our study, we used logistic regression (i.e., a typology of the generalized linear model) to compare the results of the assays performed for each of the four selected bioinsecticides (i.e., Btk and three EPNs), considering simultaneously three explicative variables, i.e., temperature condition, concentration, and time. This statistical approach is highly recommended in cases with binomial data, such as the number of SWD dead larvae out of the total larvae tested, instead of the most commonly used linear models, such as analysis of variance (ANOVA), in both biological and ecological studies [48].

The data obtained showed that Btk is highly effective, both in constant (98% SWD mortality rate at the highest concentration after 48 h of incubation) and fluctuating temperature conditions (85% and 92% for DTF_L_ and DTF_H_, respectively); however, at low DTF the efficacy is slightly reduced. Our data agree with Wraight et al. [55], who observed lower susceptibility of *Aedes stimulans* after treatment with Bt at decreasing temperatures. Comparable results were obtained by Yilmaz et al. [56]: when Bt was administered to the pine processionary (*Thaumetopoea wilkinsoni*), its efficacy was good at temperatures above 15 °C. In addition, Lysyk and Selinger [57] and Matter and Gesraha [58] reported high efficacy of several *Bacillus* isolates, including Btk, on *Stomoxys calcitrans* and *Spodoptera littoralis,* in constant temperature assays in the range of 15 to 30 °C.

The use of EPNs as pest biocontrol agents has been well demonstrated; however, several studies have indicated that variations in environmental conditions strongly influence the survival and efficacy of EPNs [59,60,61,62]. Therefore, the sensitivity of EPNs to environmental stresses, such as changes in temperature, desiccation, or increased rainfall, can drastically reduce their use as effective bioinsecticides.

In this study, we observed that the three EPN species (Sc, Sf, and Hb) showed variable efficacy on SWD larvae under the different thermal conditions chosen, accounting for their optimal temperature range of use. As previously observed [40,41,42,63], the species leading to the highest SWD lethality rate was *S. carpocapsae* (Sc) which, at constant temperature and the highest concentration, induced a mortality rate of 93%. However, when administered under DTF conditions, a reduction in its activity was detected. While a moderate decrease (75% mortality rate) was observed under the DTF_H_ condition, a significant reduction in lethality rate, which dropped to 33%, was observed under the DTF_L_ condition. The efficacy of Sc on SWD was confirmed by Cuthbertson and Audsley [64], who reported high mortality of SWD larvae in constant-temperature trials, protracted to longer treatment times (14 days). Similarly, in the work of Hubner et al. [65], high efficacy of Sc on SWD larvae was observed in tests performed at a constant temperature with administration in sandy soil; under these conditions, the authors detected an adult emergence rate of less than 20%.

Our data obtained with *S. feltiae* and *H. bacteriophora* are different from those reported in the literature [63,64,65]: even with high concentrations of nematodes, the mortality rate of larvae incubated with Sf at constant temperature did not exceed 43%, and with Hb 22%. Under conditions of temperature fluctuation, no changes were detected in the efficacy of Hb (DTF_H_), while the efficacy of Sf was further reduced (26% at DTF_L_ condition). Both works by Garriga et al. [63] and Cuthbertson and Audsley [64] reported SWD mortality rates above 70% for Hb and 80% for Sf, with administration times exceeding 48 h. These differences in the literature can be attributed to the different experimental conditions applied. We limited the time of administration to the first 48 h, considering that SWD larvae, feeding on the fruit flesh, cause extensive damage as early as the first hours post infestation. In addition, the type of substrate used for the tests (agar-sugar substrate) was developed trying to simulate the chemical and physical conditions of the pulp of thin-skinned fruits; it is known that the type of fruit (skin thickness and pulp firmness) can influence the behavior, penetration ability of EPNs and even the exploratory behavior of insect larvae [66].

Given that temperature fluctuations can influence both the behavior and physiology of both bioinsecticides and target larvae [20,21,25,26,27], in this study, we considered the effects resulting from the interaction between the two players involved in biocontrol. However, to gain an in-depth picture of their response to temperature alterations occurring in many current climate scenarios, we are investigating (work in progress) the effects of DTFs on the modulation of the host insect immune processes and the infection capability of organisms used as bioinsecticides, which is known to be key factors in the success of biocontrol [67,68].

## 5. Conclusions

The main goal of our study was to investigate the interaction between entomopathogens and SWD larvae under different thermal conditions simulating daily temperature fluctuations (DTFs). Our data showed that Btk is not affected by temperature fluctuations in both the high (DTF_H_) and low (DTF_L_) ranges (17–33 and 11–22 °C, respectively) compared to the constant temperature (25 °C). Otherwise, entomopathogenic nematodes, particularly Sc, even if effective on SWD, are susceptible to DTFs, mainly DTF_L_. Tests with Sf and Hb showed poor efficacy on SWD: even in the optimal temperature range of use indicated for these species, we detected limited activity.

Considering the drastic variations in environmental parameters linked to climate change, it is important to review the efficacy of entomopathogenic organisms and microorganisms used in the biological control of invasive species. As in our study, this could be achieved through laboratory protocols that consider the day/night temperature ranges in different geographical areas, depending on different targets and bioinsecticides.

We are aware that the results of this work are based on representative tests obtained considering some thermal ranges recorded in a temperate area of the Italian territory where SWD is particularly widespread [69]; therefore, they cannot include all environmental variables and their interactions. Nevertheless, these data can represent a starting point for further laboratory studies, which may account for both methodological and environmental variables influencing pest management in the field, thus providing preliminary indications towards a more correct and effective use of bioinsecticides.

To date, many studies on the effectiveness of pest control methods have been carried out under natural conditions in the field. 

This approach is obviously extremely important for evaluating the feasibility of the methods used and their effects on pest populations, but the ability to thoroughly control the environmental conditions, which can influence and regulate relationships between target insects and entomopathogens (Figure 6), can be achieved through laboratory trials, in which experimental parameters and environmental conditions can be finely varied and accurately monitored.

Thus, we believe that both the approach used, and the results obtained in this study can provide an important basis for understanding and revising biological control methods in the context of the drastic climatic variations currently observed.

## Figures and Tables

**Figure 1 insects-14-00043-f001:**
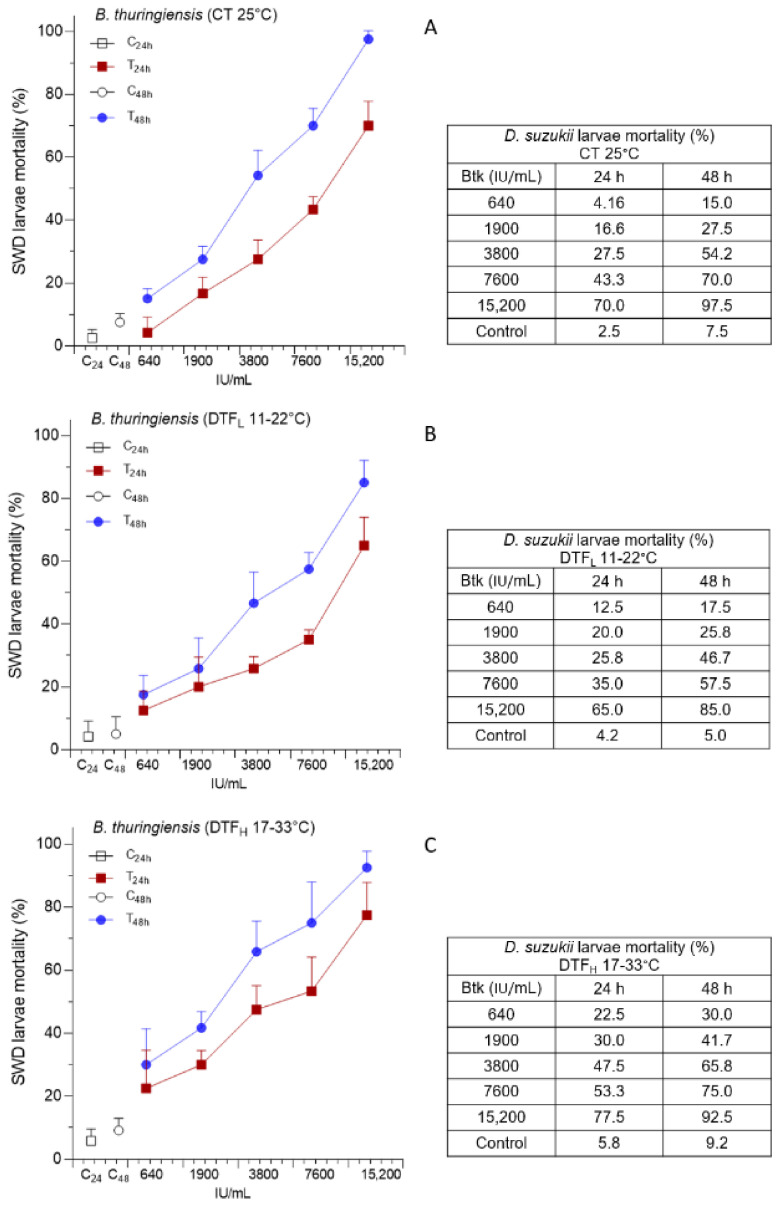
The efficacy of *Bacillus thuringiensis* var. *kurstaki* (Btk), at increasing concentrations, was evaluated as percentage mortality of Spotted Wing Drosophila (SWD) larvae (mean plus standard deviation), at 24 and 48 h, under constant temperature—CT 25 °C (**A**), low-temperature fluctuation—DTF_L_ 11–22 °C (**B**) and high-temperature fluctuation—DTF_H_ 17–33 °C (**C**).

**Figure 2 insects-14-00043-f002:**
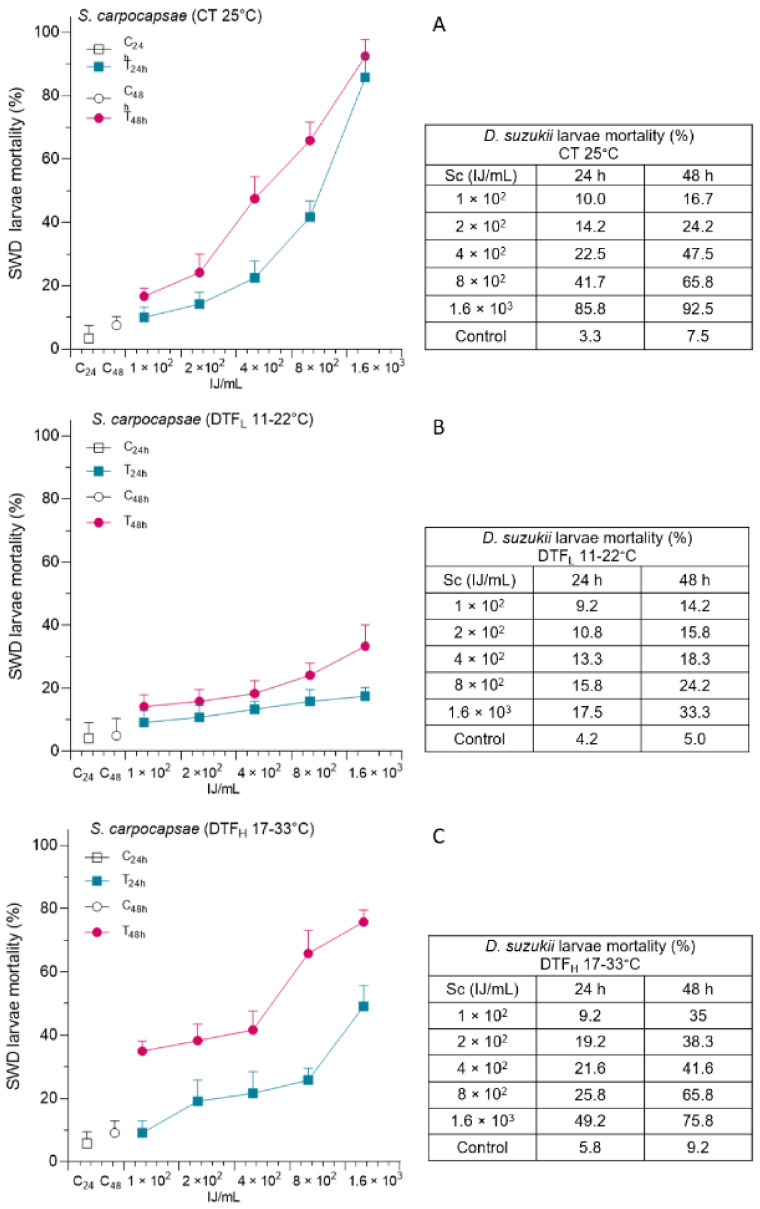
The efficacy of *Steinernema carpocapsae* (Sc), at increasing concentrations, was evaluated as percentage mortality of Spotted Wing Drosophila (SWD) larvae (mean plus standard deviation), at 24 and 48 h, under constant temperature—CT 25 °C (**A**), low-temperature fluctuation—DTF_L_ 11–22 °C (**B**) and high-temperature fluctuation—DTF_H_ 17–33 °C (**C**).

**Figure 3 insects-14-00043-f003:**
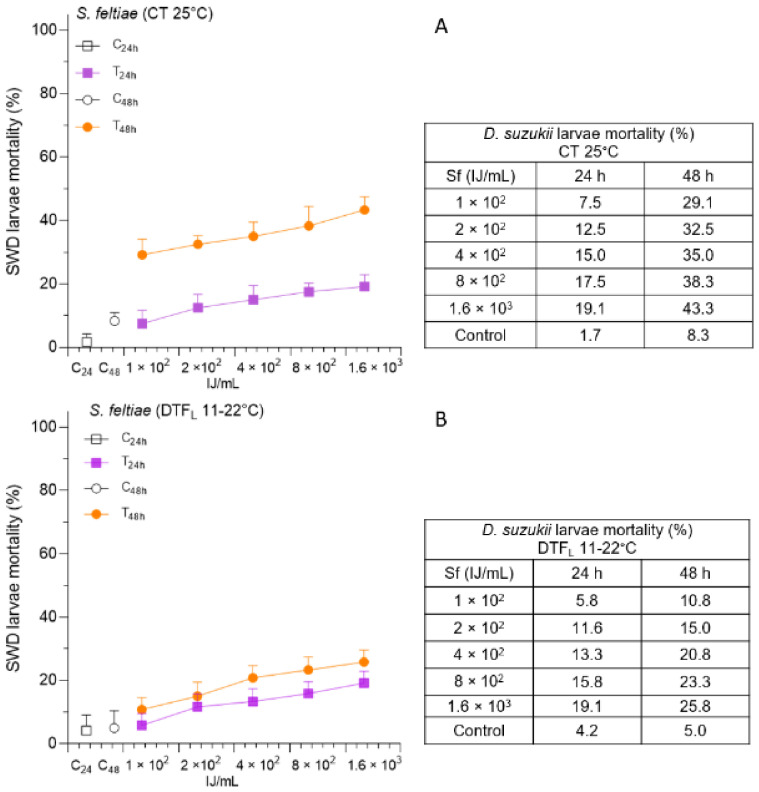
The efficacy of *Steinernema feltiae* (Sf), at increasing concentrations, was evaluated as percentage mortality of Spotted Wing Drosophila (SWD) larvae (mean plus standard deviation), at 24 and 48 h, under constant temperature—CT 25 °C (**A**), and low-temperature fluctuation—DTF_L_ 11–22 °C (**B**).

**Figure 4 insects-14-00043-f004:**
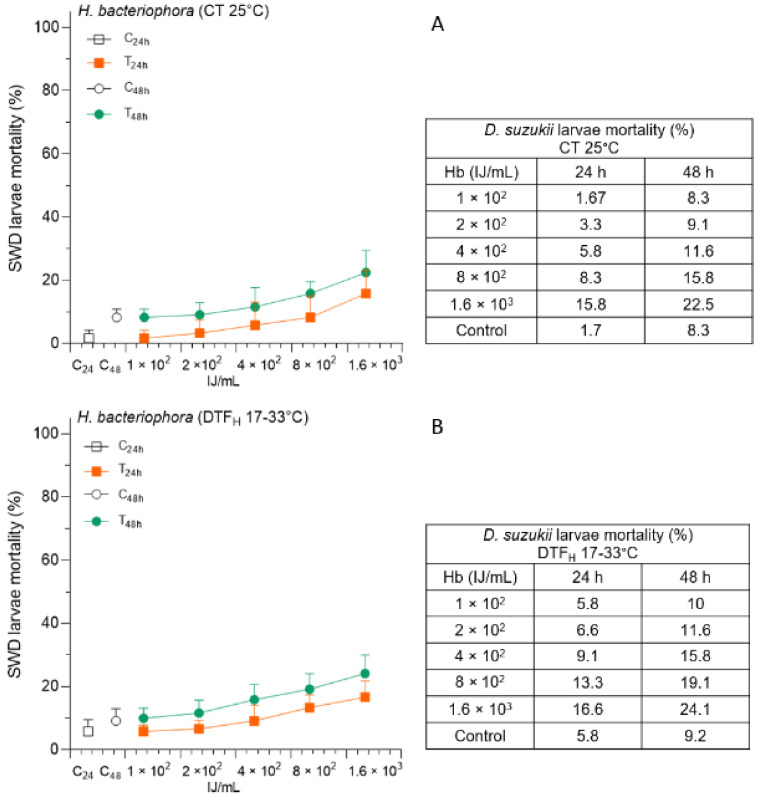
The efficacy of *Heterorhabditis bacteriophora* (Hb), at increasing concentrations, was evaluated as percentage mortality of Spotted Wing Drosophila (SWD) larvae (mean plus standard deviation), at 24 and 48 h, under constant temperature—CT 25 °C (**A**), and high-temperature fluctuation—DTF_H_ 17–33 °C (**B**).

**Figure 5 insects-14-00043-f005:**
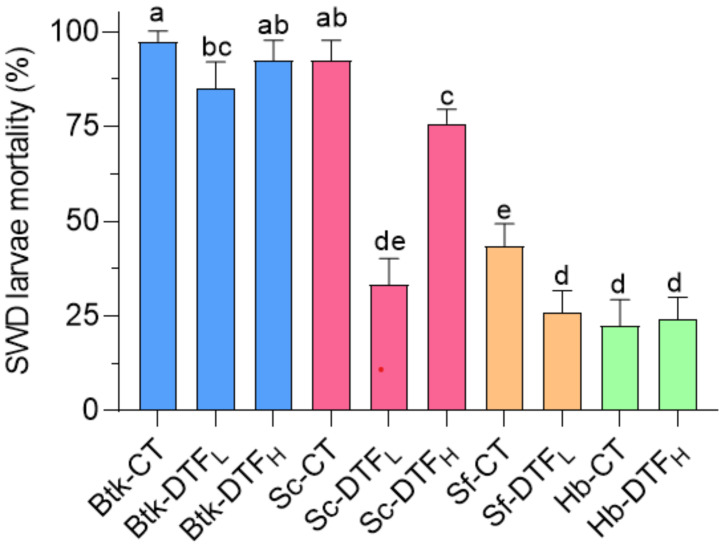
Comparison of *D. suzukii* mortality (mean plus standard deviation) caused by the four considered bioinsecticides at the highest concentration tested at 48 h at different temperature conditions (Btk-CT: *B. thuringiensis* var. *kurstaki* at a constant temperature, Btk-DTF_L_: *B. thuringiensis* var. *kurstaki* under low-temperature variation, Btk-DTF_H_: *B. thuringiensis* var. *kurstaki* under high-temperature variation; Sc-CT: *S. carpocapsae* at a constant temperature, Sc-DTF_L_: *S. carpocapsae* under low-temperature variation, Sc-DTF_H_: *S. carpocapsae* under high-temperature variation; Sf-CT: *S. feltiae* at a constant temperature, Sf-DTF_L_: *S. feltiae* under low-temperature variation; Hb-CT: *H. bacteriophora* at a constant temperature, Hb-DTF_H_: *H. bacteriophora* under high-temperature variation). Different letters above the bars indicate significant differences between estimated marginal means (*p* < 0.05).

**Figure 6 insects-14-00043-f006:**
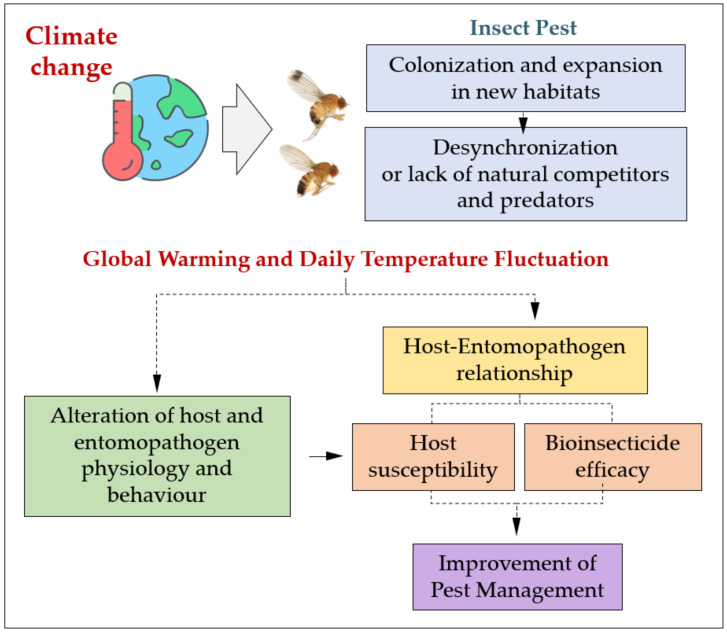
Impact of climate change on pest spread and effects of daily temperature fluctuation and global warming on biological control: pest management could be revised and improved by considering the physiological alterations induced by temperature variations resulting from climate change on bioinsecticides and targets.

**Table 1 insects-14-00043-t001:** Results of the logistic regression (logit model) carried out to compare SWD mortality at different temperature conditions (Temp), concentrations (Conc), and times (Time) for each bioinsecticide (Btk: *B. thuringiensis* var. *kurstaki*, Sc: *S. carpocapsae*, Sf: *S. feltiae*, Hb: *H. bacteriophora*). Statistics about the interactions between each couple of explicative variables, as well as between all three variables are also reported and indicated using *. Pairwise comparisons are reported in the Appendix A. DF = degrees of freedom.

		DF	χ^2^	*p*			DF	χ^2^	*p*
Btk	Logistic regression	35	905	<0.0001	Sf	Logistic regression	23	196	<0.0001
	Temperature	2	90	<0.0001		Temperature	1	86	<0.0001
	Concentration	5	1268	<0.0001		Concentration	5	191	<0.0001
	Time	1	232	<0.0001		Time	1	274	<0.0001
	Temp*Conc	10	158	<0.0001		Temp*Conc	5	104	<0.0001
	Temp*Time	2	110	<0.0001		Temp*Time	1	161	<0.0001
	Conc*Time	5	103	<0.0001		Conc*Time	5	31	<0.0001
	Temp*Conc*Time	10	74	<0.0001		Temp*Conc*Time	5	41	<0.0001
Sc	Logistic regression	35	793	<0.0001	Hb	Logistic regression	23	88	<0.0001
	Temperature	2	40	<0.0001		Temperature	1	113	<0.0001
	Concentration	5	1487	<0.0001		Concentration	5	236	<0.0001
	Time	1	116	<0.0001		Time	1	189	<0.0001
	Temp*Conc	10	701	<0.0001		Temp*Conc	5	64	<0.0001
	Temp*Time	2	42	<0.0001		Temp*Time	1	87	<0.0001
	Conc*Time	5	15	0.008		Conc*Time	5	78	<0.0001
	Temp*Conc*Time	10	112	<0.0001		Temp*Conc*Time	5	56	<0.0001

## Data Availability

Data will be available upon reasonable request to authors.

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
