# Peer review of "The Influence of Daily Temperature Fluctuation on the Efficacy of Bioinsecticides on Spotted Wing Drosophila Larvae"

_insects, 2022, doi:10.3390/insects14010043_

Round 1

Reviewer 1 Report

*Reviewer’s summary*

The study by Mastore et al examines the effects of temperature fluctuations on the efficacy of pest fruit fly management. /Drosophila suzukii/, lays eggs and feeds on ripe fruits, causing damage to agricultural producers. Bioinsecticides, such as bacteria and nematodes, provide a safer alternative to chemical insecticides to control /D. suzukii/ population. The authors used well-established bioinsecticide species to test how fluctuations in daily temperatures affect their efficacy. The study follows a simple and straightforward design, has clearly presented results, and overall excellently written. The authors reach the conclusion that some bioinsecticide species are susceptible to daily temperature changes and perform notably worse at fluctuating temperatures. The results of this study can be potentially of interest to ecologists and those involved in practical aspects of agricultural pest management.

*Major points:*

Reviewer does not have any major concerns about this study.

*Minor points:*

1.IU and IJ units need to be deciphered/explained /pro forma/.

2.In addition to describing the kinetics of temperature changes in the environmental chambers, it would be beneficial to include some graphics depicting the actual temperature in the chambers for a 24-h period. It can be provided in the supplemental data. 

Author Response

Referee 1

*Reviewer’s summary*

The study by Mastore et al examines the effects of temperature fluctuations on the efficacy of pest fruit fly management. /Drosophila suzukii/, lays eggs and feeds on ripe fruits, causing damage to agricultural producers. Bioinsecticides, such as bacteria and nematodes, provide a safer alternative to chemical insecticides to control /D. suzukii/ population. The authors used well-established bioinsecticide species to test how fluctuations in daily temperatures affect their efficacy. The study follows a simple and straightforward design, has clearly presented results, and overall, excellently written. The authors reach the conclusion that some bioinsecticide species are susceptible to daily temperature changes and perform notably worse at fluctuating temperatures. The results of this study can be potentially of interest to ecologists and those involved in practical aspects of agricultural pest management.

Major points:

Reviewer does not have any major concerns about this study.

Thank You very much for your positive comment.

Minor points:

1.IU and IJ units need to be deciphered/explained /pro forma/.

Acronyms were described in the text (row 191 for IU and row 159 for IJ)

2.In addition to describing the kinetics of temperature changes in the environmental chambers, it would be beneficial to include some graphics depicting the actual temperature in the chambers for a 24-h period. It can be provided in the supplemental data. 

As required, we included a graph with temperature variation set in the climatic chamber for the two DTF conditions (see Fig. SM1 In the supplementary material).

Reviewer 2 Report

Brief Summary:

This paper highlights the challenges of insect pest management in these modern times. Climate change is expanding the habitable range of many insect pests into new areas that were previously too cold for the pests to survive in. This results in an overall increase of chemical pesticide use which causes even more environmental damage than seen in previous decades. The alternative strategy of bioinsecticides as an environmentally friendly and sustainable strategy is very promising; however, given that bioinsecticides are live agents themselves they are susceptible to fluctuating temperatures and weather which can dramatically impact their efficacy, unlike chemical pesticides. The authors decide to study the effects of fluctuating temperatures on the efficacy of common bioincesticides to infect and destroy the fruit fly pest, Drosophila suzukii, to determine which bioinsecticide is most effective across a wide range of temperatures and which alternatives might be promising in a smaller subset of temperatures (more stable climes).

General Concepts Comments:

In the Materials and methods section 2.2 (lines 122-127), it is not clear where the D. suzukii lines came from. Are they laboratory strains from personal labs or stock centers? Were they captured from local fields? Are the strains isogenic or are they likely to harbor a diverse array of single nucleotide polymorphisms (SNPs) among the populations being reared for this study? The reason why I point this out is that there have been quite a few papers utilizing the Drosophila melanogaster DGRP Genetic Reference Panels and similar resources to identify isogenic lines with sequenced SNPs that affect phenotypes through canonical and cryptic genetic variation. As far as I am aware, there are no equivalent resources for Drosophila suzukii, but it is still important to identify if it was a laboratory strain or not, since they tend to be inbred and more similar to each other than in natural populations, which could potentially influence your results here. I do wish to make clear here that I am not asking for these experiments to be repeated with a laboratory isogenic strain if that is not the case, only that the conditions and origin of the lines be identified here and expanded upon in the discussion as possible limitations of the present study and future aims as part of examining immune response in D. suzukiii along with the bioinsecticide agents.

I agree with the authors that the laboratory setup provides the best conditions for testability of the DTF hypothesis of bioinsecticide efficacy that would be more difficult to control for through capture and monitoring of natural populations affecting nearby fruit groves and fields. The methods in this paper outline the many steps to control independent variables as much as possible, such as age of the larvae/nematodes, the number of individual bioincesticide agents, and so on. Combined with robust statistical tests, these methods are appropriate for testing the DTF hypothesis on bioinsecticide efficacy.

The only issue I would like to raise here with the methods documentation is the delay in mentioning the replicates. When reading the Materials and Methods section 2.3 Biocontrol Assays section (starting at line 143), I did not see any mention of replication here and it prompted me to wonder how many replicates were done, if any. My question was answered when I examined section 2.4 "Data analyses" where the authors mentioned six replicates (line 175). I recommend mentioning the replicates much earlier in the Biocontrol Assay to limit confusion about reproducibility.

Specific Comments:

In terms of the writing of the manuscript, it is very clear and readable, with the exception of several minor spelling mistakes that can easily be caught by proofreaders. There are a couple of passages that was unclear to me specifically and wish to share here:

1)  lines 43-44: In the first sentence of the Introduction section, "Greenhouse gases derived from human activities are the main responsible for...." is missing a term between 'responsible' and 'for'. This is a minor grammar issue under most circumstances but considering that this is the first line of the Introduction, I wanted to make sure that this was caught since most readers will be reading this to see if they want to continue reading the paper.

2)  lines 157-158: I did not see any indication of what RH means and might be confusing for readers. I recommend spelling it out the first time you refer to RH.

3)  lines 345-347: The sentence "Either 345 Garriga et al. [58] or Cuthbertson and Audsley [59] reported instead SWD mortality rates 346 above 70% for Hb and 80% for Sf, in assays longer than 48 hours" was confusing to read and understand who did what in these previous reports. A suggestion for rephrasing this might be something like "Garriga et al and Cuthbertson/Audsley reported SWD mortality rates above __% for Hb and __% for Sf, respectively, in assays exceeding 48 hours."

Manuscript Comments:

Overall, I think the manuscript is very clear and relevant to several fields, including pest management, ecology, and host-pathogen interactions. The majority of the references cited within this manuscript are from recent publications and are not excessive in self-citations. The number of references is also not excessive, yet they provide a snapshot of the diversity of previous publications that the authors read to construct their hypotheses here. However, I would strongly suggest that the authors incorporate the following reference in their paper: "Species-specific modulation of food-search behavior by respiration and chemosensation in Drosophila larvae" (https://doi.org/10.7554/eLife.27057). The paper researches the food-search behaviors of D. melanogaster and D. suzukii and finds that surface hardness affects larvae exploratory behavior. Given that the authors took care to mimic the conditions of thin-skinned fruits infected by D. suzukii (lines 157-159), including this reference should elevate the importance of this step in terms of controlling for experimental variables.

I find this manuscript to be scientifically sound with a robust experimental design that facilitated the fundamental discoveries of DTFs on bioincesticide efficacy. The materials and methods section are sufficiently detailed that other researchers should have little issue setting up their own experiments in the same manner, which will aid in reproducibility of results as reported in this manuscript. The majority of the figures are graphs with accompanying data tables and are very easy to interpret the effect of fluctuating temperatures on bioinsecticide efficacy of D. suzukii larvae. Figure 5 does a nice job aggregating the data from the previous figures all together to make it easy to compare D. suzukii mortality rates with the highest concentrations of each bioinsecticide surveyed. Figure 6 is a very nice graphical abstract that visualizes the potential impacts of climate change of insect pest invasion of new habitats and how climate impact might affect organismal and ecological processes at different levels.

The inclusion of Table 1 and supplementary tables allows readers to examine the standard deviation, degrees of freedom, and other attributes that were utilized to calculate the p-value to determine statistical significance of the results and were applied consistently across the board. The conclusions are consistent with the reported results and authors provided a reasonable explanation for the deviation of Hb and Sf efficacy in comparison to previous reports (line 347). I do have a suggestion for the authors to expand more on the reasons for using the various statistical tests in the discussion section. There is a bit of this detail in the Materials and Methods, but I predict that the readership will be diverse for this paper due to the various disciplines it covers (climate change, ecology, pest management, entomology, etc) and I feel that it would benefit this diverse readership who may not be as well-versed in biostatistics to understand more about why these particular tests were used (and not basic ones like t-tests, etc)

The authors declare that they have no conflicts of interest, and that data will be made available upon reasonable request. These statements satisfy the requirements of the journal.

Author Response

Referee 2

This paper highlights the challenges of insect pest management in these modern times. Climate change is expanding the habitable range of many insect pests into new areas that were previously too cold for the pests to survive in. This results in an overall increase of chemical pesticide use which causes even more environmental damage than seen in previous decades. The alternative strategy of bioinsecticides as an environmentally friendly and sustainable strategy is very promising; however, given that bioinsecticides are live agents themselves they are susceptible to fluctuating temperatures and weather which can dramatically impact their efficacy, unlike chemical pesticides. The authors decide to study the effects of fluctuating temperatures on the efficacy of common bioincesticides to infect and destroy the fruit fly pest, Drosophila suzukii, to determine which bioinsecticide is most effective across a wide range of temperatures and which alternatives might be promising in a smaller subset of temperatures (more stable climes).

General Concepts Comments:

In the Materials and methods section 2.2 (lines 122-127), it is not clear where the D. suzukii lines came from. Are they laboratory strains from personal labs or stock centers? Were they captured from local fields? Are the strains isogenic or are they likely to harbor a diverse array of single nucleotide polymorphisms (SNPs) among the populations being reared for this study? The reason why I point this out is that there have been quite a few papers utilizing the Drosophila melanogaster DGRP Genetic Reference Panels and similar resources to identify isogenic lines with sequenced SNPs that affect phenotypes through canonical and cryptic genetic variation. As far as I am aware, there are no equivalent resources for Drosophila suzukii, but it is still  important to identify if it was a laboratory strain or not, since they tend to be inbred and more similar to each other than in natural populations, which could potentially influence your results here. I do wish to make clear here that I am not asking for these experiments to be repeated with a laboratory isogenic strain if that is not the case, only that the conditions and origin of the lines be identified here and expanded upon in the discussion as possible limitations of the present study and future aims as part of examining immune response in D. suzukiii along with the bioinsecticide agents.

We agree with this comment, and we have added the information in the text (rows 149-151).

“The third stage of D. suzukii larvae used for all assays was obtained from a laboratory culture of specimens collected in Catalonia (NE Spain), kindly provided by Prof. Del Pino (Autonomous University of Barcelona, UAB) in 2018.”

I agree with the authors that the laboratory setup provides the best conditions for testability of the DTF hypothesis of bioinsecticide efficacy that would be more difficult to control for through capture and monitoring of natural populations affecting nearby fruit groves and fields. The methods in this paper outline the many steps to control independent variables as much as possible, such as age of the larvae/nematodes, the number of individual bioincesticide agents, and so on. Combined with robust statistical tests, these methods are appropriate for testing the DTF hypothesis on bioinsecticide efficacy.

The only issue I would like to raise here with the methods documentation is the delay in mentioning the replicates. When reading the Materials and Methods section 2.3 Biocontrol Assays section (starting at line 143), I did not see any mention of replication here and it prompted me to wonder how many replicates were done, if any. My question was answered when I examined section 2.4 "Data analyses" where the authors mentioned six replicates (line 175). I recommend mentioning the replicates much earlier in the Biocontrol Assay to limit confusion about reproducibility.

We mentioned the number of replicates also in the subchapter 2.3 Biocontrol Assays (row 180).

Specific Comments:

In terms of the writing of the manuscript, it is very clear and readable, with the exception of several minor spelling mistakes that can easily be caught by proof readers. There are a couple of passages that was unclear to me specifically and wish to share here:

1) lines 43-44: In the first sentence of the Introduction section, "Greenhouse gases derived from human activities are the main responsible for...." is missing a term between 'responsible' and 'for'. This is a minor grammar issue under most circumstances but considering that this is the first line of the Introduction, I wanted to make sure that this was caught since most readers will be reading this to see if they want to continue reading the paper.

We reworded the sentence (rows 45-47).

2) lines 157-158: I did not see any indication of what RH means and might be confusing for readers. I recommend spelling it out the first time you refer to RH.

We mentioned it at row 154.

3) lines 345-347: The sentence "Either 345 Garriga et al. [58] or Cuthbertson and Audsley [59] reported instead SWD mortality rates above 70% for Hb and 80% for Sf, in assays longer than 48 hours" was confusing to read and understand who did what in these previous reports. A suggestion for rephrasing this might be something like "Garriga et al and Cuthbertson/Audsley reported SWD mortality rates above __% for Hb and __% for Sf, respectively, in assays exceeding 48 hours."

We clarified the sentence (rows 388-390).

Manuscript Comments:

Overall, I think the manuscript is very clear and relevant to several fields, including pest management, ecology, and host-pathogen interactions. The majority of the references cited within this manuscript are from recent publications and are not excessive in self-citations. The number of references is also not excessive, yet they provide a snapshot of the diversity of previous publications that the authors read to construct their hypotheses here. However, I would strongly suggest that the authors incorporate the following reference in their paper: "Species-specific modulation of food-search behavior by respiration and chemosensation in Drosophila larvae"(https://doi.org/10.7554/eLife.27057). The paper researches the food-search behaviors of D. melanogaster and D. suzukii and finds that surface hardness affects larvae exploratory behavior. Given that the authors took care to mimic the conditions of thin-skinned fruits infected by D. suzukii (lines 157-159), including this reference should elevate the importance of this step in terms of controlling for experimental variables.

We thank for the positive comments and the interesting suggestion. We included the reference in the discussion section (line 398).

I find this manuscript to be scientifically sound with a robust experimental design that facilitated the fundamental discoveries of DTFs on bioincesticide efficacy. The materials and methods section are sufficiently detailed that other researchers should have little issue setting up their own experiments in the same manner, which will aid in reproducibility of results as reported in this manuscript. The majority of the figures are graphs with accompanying data tables and are very easy to interpret the effect of fluctuating temperatures on bioinsecticide efficacy of D. suzukii larvae. Figure 5 does a nice job aggregating the data from the previous figures all together to make it easy to compare D. suzukii mortality rates with the highest concentrations of each bioinsecticide surveyed. Figure 6 is a very nice graphical abstract that visualizes the potential impacts of climate change of insect pest invasion of new habitats and how climate impact might affect organismal and ecological processes at different levels.

Many thanks for the appreciation of our work.

The inclusion of Table 1 and supplementary tables allows readers to examine the standard deviation, degrees of freedom, and other attributes that were utilized to calculate the p-value to determine statistical significance of the results and were applied consistently across the board. The conclusions are consistent with the reported results and authors provided a reasonable explanation for the deviation of Hb and Sf efficacy in comparison to previous reports (line 347). I do have a suggestion for the authors to expand more on the reasons for using the various statistical tests in the discussion section. There is a bit of this detail in the Materials and Methods, but I predict that the readership will be diverse for this paper due to the various disciplines it covers (climate change, ecology, pest management, entomology, etc) and I feel that it would benefit this diverse readership who may not be as well-versed in biostatistics to understand more about why these particular tests were used(and not basic ones like t-tests, etc)

We thank you also for this suggestion. We changed ANOVA with logistic regression considering the useful comment provided by Referee 3. We included in the discussion section the reason for using the new statistical approach (rows 349-355).

In our study, we used logistic regression (i.e., a typology of generalized linear model) to compare the results of the assays performed for each of the four selected bioinsecticides (i.e., Btk and three EPNs), considering simultaneously three explicative variables, i.e., temperature condition, concentration, and time. This statistical approach is highly recommended in case of binomial data, such as the number of SWD dead larvae out of the total larvae tested, instead of the most commonly used linear models, such as analysis of variance (ANOVA), in both biological and ecological studies [50].”

The authors declare that they have no conflicts of interest, and that data will be made available upon reasonable request. These statements satisfy the requirements of the journal.

Reviewer 3 Report

This study by Mastore and colleagues provides some original insights aimed at assessing the effects of differing constant and fluctuating temperature profiles on the efficacy of bioinsecticides against SWD larvae. I recommend it for publication, but after a major revision that should solve the following manuscript’s constraints:

1) The introduction and discussion provide no insight on how this MS relates to the various other ones cited in the text or concerns that have been raised by other researchers. This article should provide details on all these fronts to provide the proper context for the work. Authors do not present any hypotheses or expectations that could be connected to previous studies; adding these details will improve the paper. This article should provide details on all these fronts to provide the proper context for the work.

2) Some of the authors statements (e.g., Lns:70-73 & Lns:74-80) would be much stronger if they tie their work to the body of literature that has built up on the bioecology and reproductive biology of mass-produced endo- and ectoparasite insect biocontrol agents (BCAs) for field releases in CBC programs (see my comments under #4). They all point to the same direction and could be connected to this study. Some examples are J. Econ. Entomol. 112: 1560-1574 (mass produced ectoparasite BCAs) or J. Econ. Entomol. 112:1062-1072 (mass produced endoparasite BCAs), but there are others too. These studies provide strong evidence of increased longevity in BCAs reared at non-stressful low temperatures when compared to higher temperature regimes. They further suggest that the parasitism or percentage host mortality was significantly higher at intermediate temperatures than at cline margins. Adding these details will improve the paper.

3) Unjustified/inappropriate analyses and/or flawed reasoning: Lns:172-190: ANOVA/ANCOVA on percentage mortality data? If so, consider an analysis better suited to this response variable, logistic GLM is one, binomial distribution is another one (see Warton & Hui’s MS on asinine arcsine transformations of proportions data published in Ecology 2011 92, 3-10). The major difficulty with modelling proportion data is that the responses are strictly bounded. There is no way that the percentage dying can be greater than 100% or less than 0%. But if we use simple techniques such as regression, analysis of variance or covariance, then the fitted model could quite easily predict negative values or values greater than 100%, especially if the variance was high and many of the data were close to 0 or close to 100%. The logistic curve is commonly used to describe data on proportions, because, unlike the straight-line model, it asymptotes at 0 and 1 so that negative proportions and responses of more than 100% cannot be predicted. Briefly, proportions are based on number of cases. Would you give the same weight to a proportion of 2 out of 4 cases (not very reliable) and a more reliable proportion of 20 out of 40 cases? The natural solution is to use the odds and odds ratio, and a binomial distribution to test for change in proportion as a change in the odds, as described in the arcsine asinine publication (see Ecology 2011 92, 3-10). That way you give 50 % of 40 its due, compared to 50% of 4. Results and Discussion sections should be revised accordingly.

4) My other concern is that the authors are extrapolating the applicability of their results beyond what the design supports. These are only development data from a single constant temperature profile (25C) and a couple of sets of highly artificial fluctuating temperature profiles (i.e., 11-22C and 17-33) that changed over 6h, so the inference power of the paper is very limited, but authors do not acknowledge this detail at all and need to be more forthcoming. This is a critical limitation of the study, and the authors must concede and discuss this. It is well known that most of laboratory experiments are conducted under constant temperatures whereas in nature daily temperature fluctuations can be very wide. The interaction of daily cyclic temperatures that fluctuate over 24h with nonlinear efficacy of BCAs can introduce significant deviations from the parameters developed here (see examples from J. Econ. Entomol. 2019, 112: 1560-1574 and J. Econ. Entomol. 2019, 112:1062-1072 mentioned above). But this has not been investigated in this study. So, I am suggesting to the authors to tone-down the language a little and admit that there are still substantive uncertainties to be considered, including uncertainty as to how generalizable the results are to open field conditions.

The next draft of this paper will need to be dramatically different to have a chance at publication in my humble opinion. This is not to diminish the data gathered in this study, they are of value. But it is important for the authors not to overgeneralize, and to warn the reader, including regulatory agencies, against doing so as well. Good luck!

Author Response

Referee 3 (see changes in the manuscript highlighted in light blue)

This study by Mastore and colleagues provides some original insights aimed at assessing the effects of differing constant and fluctuating temperature profiles on the efficacy of bioinsecticides against SWD larvae. I recommend it for publication, but after a major revision that should solve the following manuscript’s constraints:

1) The introduction and discussion provide no insight on how this MS relates to the various other ones cited in the text or concerns that have been raised by other researchers. This article should provide details on all these fronts to provide the proper context for the work. Authors do not present any hypotheses or expectations that could be connected to previous studies; adding these details will improve the paper. This article should provide details on all these fronts to provide the proper context for the work.

We tried to better contextualize our work in the introduction (rows 59-72).

2) Some of the authors statements (e.g., Lns:70-73 & Lns:74-80) would be much stronger if they tie their work to the body of literature that has built up on the bioecology and reproductive biology of mass-produced endo- and ectoparasite insect biocontrol agents (BCAs) for field releases in CBC programs (see my comments under #4). They all point to the same direction and could be connected to this study. Some examples are J. Econ. Entomol. 112: 1560-1574 (mass produced ectoparasite BCAs) or J. Econ. Entomol. 112:1062-1072 (mass produced endoparasite BCAs), but there are others too. These studies provide strong evidence of increased longevity in BCAs reared at non-stressful low temperatures when compared to higher temperature regimes. They further suggest that the parasitism or percentage host mortality was significantly higher at intermediate temperatures than at cline margins. Adding these details will improve the paper.

We thank you for the interesting suggestion, and we included the references in the text (ref. 12-16, 22-23).

3) Unjustified/inappropriate analyses and/or flawed reasoning: Lns:172-190: ANOVA/ANCOVA on percentage mortality data? If so, consider an analysis better suited to this response variable, logistic GLM is one, binomial distribution is another one (see Warton & Hui’s MS on asinine arcsine transformations of proportions data published in Ecology 2011 92, 3-10). The major difficulty with modelling proportion data is that the responses are strictly bounded. There is no way that the percentage dying can be greater than 100% or less than 0%. But if we use simple techniques such as regression, analysis of variance or covariance, then the fitted model could quite easily predict negative values or values greater than 100%, especially if the variance was high and many of the data were close to 0 or close to 100%. The logistic curve is commonly used to describe data on proportions, because, unlike the straight-line model, it asymptotes at 0 and 1 so that negative proportions and responses of more than 100% cannot be predicted. Briefly, proportions are based on number of cases. Would you give the same weight to a proportion of 2 out of 4 cases (not very reliable) and a more reliable proportion of 20 out of 40 cases? The natural solution is to use the odds and odds ratio, and a binomial distribution to test for change in proportion as a change in the odds, as described in the arcsine asinine publication (see Ecology 2011 92, 3-10). That way you give 50 % of 40 its due, compared to 50% of 4. Results and Discussion sections should be revised accordingly.

Thank you very much for this comment. We carefully read the papers you suggested and modified statistical analyses accordingly. Specifically, we used the approach proposed by Warton and Hui (ref. 50) to binomial data. As reported in the “data analyses” subchapter 2.4, we used logistic regression (logit model) instead of ANOVA after arcsine square root transformation of data. Related results and discussion were modified accordingly. Also, table 1 and all supplementary tables were corrected.

4) My other concern is that the authors are extrapolating the applicability of their results beyond what the design supports. These are only development data from a single constant temperature profile (25C) and a couple of sets of highly artificial fluctuating temperature profiles (i.e., 11-22C and 17-33) that changed over 6h, so the inference power of the paper is very limited, but authors do not acknowledge this detail at all and need to be more forthcoming. This is a critical limitation of the study, and the authors must concede and discuss this. It is well known that most of laboratory experiments are conducted under constant temperatures whereas in nature daily temperature fluctuations can be very wide. The interaction of daily cyclic temperatures that fluctuate over 24h with nonlinear efficacy of BCAs can introduce significant deviations from the parameters developed here (see examples from J. Econ. Entomol. 2019,112: 1560-1574 and J. Econ. Entomol. 2019, 112:1062-1072 mentioned above). But this has not been investigated in this study. So, I am suggesting to the authors to tone-down the language a little and admit that there are still substantive uncertainties to be considered, including uncertainty as to how generalizable the results are to open field conditions.

We highlighted both the limits of our work and the potentialities of similar studies in the introduction and conclusions section (rows 59-72, 127-133 and 421-427).

The next draft of this paper will need to be dramatically different to have a chance at publication in my humble opinion. This is not to diminish the data gathered in this study, they are of value. But it is important for the authors not to overgeneralize, and to warn the reader, including regulatory agencies, against doing so as well. Good luck!

 We hope that the revised version of the manuscript, which now includes your suggestions will meet the standards for publication in Insects.

Round 2

Reviewer 3 Report

Authors have done a fine job addressing all of my original concerns and those of other reviewers. I have no further comments to improve the manuscript. Thank you!